# Calcium Transport along the Axial Canal in *Acropora*

**Yixin Li** [1] , **Xin Liao** [2] , **Chunpeng He** [1],* **and Zuhong Lu** [1],*

1   State Key Laboratory of Bioelectronics, School of Biological Science and Medical Engineering, Southeast University, Nanjing 210096, China; 220174582@seu.edu.cn
2   Guangxi Key Laboratory of Mangrove Conservation and Utilization, Guangxi Mangrove Research Center, Guangxi Academy of Sciences, Beihai 536000, China; liaox@mangrove.org.cn
*   Correspondence: cphe@seu.edu.cn (C.H.); zhlu@seu.edu.cn (Z.L.); Tel.: +86-177-9855-0441 (C.H.); +86-186-5180-0622 (Z.L.)

**Abstract:** In *Acropora*, the complex canals in a coral colony connect all polyps to a holistic network, enabling them to collaborate in performing biological processes. There are various types of canals, including calice, axial canals, and other internal canals, with structures that are dynamically altered during different coral growth states due to internal calcium transport. In this study, we investigated the morphological changes in the corallite of six *Acropora muricata* samples by high resolution micro-computed tomography, observing the patterns of calcium carbonate deposition within axial corallite during processes of new branch formation and truncated tip repair. We visualized the formation of a new branch from a calice and the calcium carbonate deposition in the axial canal. Furthermore, the diameter and volume changes of the axial canal in truncated branches during rebuilding processes were calculated, revealing that the volume ratio of calcareous deposits in the axial canal exhibit significant increases within the first three weeks, returning to levels in the initial state in the following week. This work demonstrates that calcium carbonate can be stored temporarily and then remobilized as needed for rapid growth. The results of this study shed light on the control of calcium carbonate deposition and growth of the axial corallite in *Acropora*.

**Keywords:** axial canal; reef-building coral; high-resolution micro-computed tomography; *Acropora muricata*; calcium transport; deposit

## 1. Introduction

Coral reefs are highly diverse ecosystems characterized by reef-building corals [1,2]. Reef-building corals are essential for the maintenance of the biodiversity and ecological functioning of coral reefs [3–6]. Among the major reef-building corals, *Acropora* species are responsible for forming the immense calcium carbonate substructure, which is the core of a reef and supports its thin living skin [7]. The tissue gastrovascular canals lie within the lumen of the skeleton in an *Acropora* colony, and materials can be transported within the canals [8,9]. The complex canals in a colony connect all polyps into a holistic network to collaborate in performing biological processes [10]. Among these processes, the biomineralization carried out in coral polyps deserves attention, as it can sequester carbon and is involved in reef formation [11–15]. All polyps in the *Acropora* colony mineralize carbonate or induce calcareous precipitation, and calcium can be carried over considerable distances inside the coral colony toward the zones of maximum growth and calcification [16–19]. The canal network in the colony forms a non-radial symmetry transport system for calcium transport during coral growth [20]. In this network, the gastrovascular canal system consists of the axial canal lying within the axial corallite, radial canals in the lateral corallites, and a network of smaller diameter canals connecting these together [21]. The movement of fluid within the gastrovascular system can bring materials to different parts of the colony as required [22]. Former researches described temporal and spatial patterns of calcium carbonate accretion in *Acropora cervicornis*, and discussed the diel cycle as well as decadal

cycles of carbonate deposition in the axial corallite [21,23]. However, the patterns of calcium transport in the axial canal and details of its structural transformation during coral growth remain obscure, as literature on this subject is scarce [24]. Meanwhile, although the transport of organic compounds within coral colonies has been suggested in various works, most are inferences based on markers or elemental measurements [25–29], rather than direct visualizations of structural changes in coral colonies. Errors may occur in the measurements, because of the requirement to remove tissue from the skeletons [30].

The non-transparent skeleton influences direct observation of the distribution, parameters, and relationships among canals in coral colonies [20]. Experiments with traditional biological methods have provided very limited and circumstantial evidence in changes in the diameter of the axial corallite [31–33]. To solve this problem, high-resolution computed tomography (HRCT) has gained increasing attention [34,35]. HRCT can be used to non-destructively capture the morphology and internal structure of coral colonies [36,37]. Compared with traditional biological techniques, such as scanning electron microscope (SEM) and grinding sections, HRCT has multiple advantages [10]. Skeletal reconstruction through HRCT can be used directly on living corals, for which complicated and potentially destructive preparations, such as pickling or fixing, are not required [38]. HRCT can reveal the delicate internal skeletal structures in coral colonies that are easily destroyed using traditional techniques [39]. Moreover, all colony skeletal information can be captured in detail at once [40]. Any position and section in a colony can be observed as needed, saving coral resources and eliminating the burden of multiple measurements with complete sample analysis achieved in a single process [41].

In this study, we explored calcium transport in the axial canal during different physiological states of coral growth by comparing the skeleton and canal morphology between two normal growing states. One is the growth process of a new branch in a coral colony, and the other is the self-rebuilding process of truncated branches. We used HRCT to reconstruct six representative samples of *Acropora muricata*, which is common and frequently a dominant species in coral reefs. Furthermore, we calculated related parameters of the axial corallite during new branch formation and truncated branch rebuilding. The pattern regulation of axial corallite in colony formation was visualized in both of these growth states, revealing the regulatory processes of the axial corallite in the calcium transport system. Thus, calcium transport along the axial canal in *Acropora* could be determined.

This study expands our understanding of calcium transport patterns in *Acropora*, which sheds light on the control of calcium carbonate deposition and growth of the axial corallite in *Acropora* [21–23,42]. The data enable a much more detailed study of the calicoblastic tissue during the buildup of calcium carbonate and, conversely, during the dissolution of the axial corallite in the 28 day period. This is ripe for a transmission electron microscope (TEM) study of the tissue, and could contribute greatly to an understanding of the process of calcification [43].

## 2. Methods and Materials

### 2.1. Sample Collection

All six *A. muricata* samples in this study were collected from the Xisha Islands (Paracel Islands, 16°53′ N, 112°17′ E) of the South China Sea, in 2018. All samples, which occurred in large arborescent colonies forming thickets, were found in tropical shallow reefs of marine neritic, from depths of about 5 to 10 m. The daily mean temperature was between 23.2 and 29.2 °C. The coral samples were kept whole and housed in our laboratory coral tank, where all conditions were simulated to reflect those of their habitat in the South China Sea. These samples were kept in the tank for about one to three months before the HRCT test. Among these *A. muricata* samples, one was a colony (about 20 cm × 20 cm × 25 cm), and the other five were coral branches (length of about 4 cm, diameter between 0.5 and 1 cm) from different colonies.

## 2.2. Coral Culture System

Our coral samples were cultured with the laboratory auto calibration balance system [44] in a standard Red Sea® tank (redsea575, Red Sea Aquatics Ltd., London, UK), following the Berlin method. The temperature was kept at 25 °C, and the salinity (Red Sea Aquatics Ltd., London, UK) was 1.025. The culture system was maintained using a Protein Skimmer (regal250s, Honya Co. Ltd., Shenzhen, China), a water chiller (tk1000, TECO Ltd., Taiwan, China), three coral lamps (AI®, Red Sea Aquatics Ltd., London, UK), two wave devices (VorTech™ MP40, EcoTech Marine Ltd., Bethlehem, PA, USA), and a calcium reactor (Calreact 200, Honya Co. Ltd., Shenzhen, China).

Around 20 kg of live rocks, which were also collected from the South China Sea, were placed in the coral tank. These live rocks provided the structure of the growth environment and some necessary microorganisms. We also added minerals to the tank weekly, including Mg, Ca, KH, K, I, and Fe.

## 2.3. HRCT Test

We analyzed six *A. muricata* samples from the South China Sea using three dimensional models constructed with the 230 kV latest-generation X-ray microfocus-computed tomography system (Phoenix v | tome | x m, General Electric (GE)), at Yinghua NDT, Shanghai, China. Two-dimensional image reconstructions of each specimen from matrices of scan slices were assembled using proprietary software from GE. The relevant parameters are shown in Table 1.

**Table 1.** Parameters of the HRCT tests.

| Sample | Voltage | Current | Voxel Size | Timing | Number of Images | Image Width | Image Height |
|---|---|---|---|---|---|---|---|
| *Acropora* colony one | 150 kV | 180 µA | 37 µm | 1 s | 2000 | 3990 pixels | 4000 pixels |
| *Acropora* branch one | 130 kV | 60 µA | 6 µm | 500 ms | 1500 | 2800 pixels | 4000 pixels |
| *Acropora* branch two | 120 kV | 115 µA | 12 µm | 500 ms | 2400 | 1980 pixels | 2000 pixels |
| *Acropora* branch three | 130 kV | 60 µA | 6 µm | 500 ms | 1500 | 2800 pixels | 4000 pixels |
| *Acropora* branch four | 130 kV | 100 µA | 9 µm | 500 ms | 2500 | 1985 pixels | 2000 pixels |
| *Acropora* branch five | 160 kV | 70 µA | 9 µm | 500 ms | 1600 | 1500 pixels | 4000 pixels |

## 2.4. Internal Canal Reconstruction

Slice data derived from the scans were then analyzed and manipulated using VG software. The 3D reconstructions were created in Mimics (v20.0) software and VG Studio Max (v3.3.0), following the method as previously described [10]. The images of the reconstructions were exported from Mimics and VG Studio Max and finalized in Adobe Photoshop CC 2019 and Adobe Illustrator CC 2019.

## 2.5. Truncation Experiment

We selected four groups of *A. muricata* branches of a similar size (4 cm) and shape (a column with a diameter between 5 mm and 1 cm from the tip to the bottom) to truncate their tips at same position and culture them in the same environments. The sampling occurred the same time each day, at 12 o'clock noon. The illuminance in the tank was between 600 and 800 PAR (µmol/m$^2$·s), and the time of irradiation was between 8 AM and 8 PM. The truncated samples were assessed using HRCT at day 0, 14, 21, and 28 (*A. muricata* branch 2–5) for HRCT detection.

## 2.6. Calculation of the Calcareous Deposit Volume Ratio in the Axial Canal

The calculation of the calcareous deposit volume ratio in the axial canal was performed in the VG Studio Max 3.3. First, we used "surface determination" to distinguish the area of skeleton reconstruction and porosity. Then, we calculated the volume of calcareous deposits, and used the "erode/dilate" mode to select the entire area of the axial canal. After that, we used the "porosity/inclusion analysis module" to reconstruct the internal porosity for the volume calculation. Thus, with the volume of both calcareous deposits and the axial canal, we were able to obtain the calcareous deposit volume ratio in the axial canal of each sample.

## 3. Results

### 3.1. The Morphology and Internal Structure of Corallite in A. muricata

The three-dimensional skeletal structures of six *A. muricata* samples were reconstructed by HRCT, including both the surface morphology and the internal structural characteristics, allowing us to study the structural pattern of the axial corallite in the processes of coral branch formation (Figures 1 and 2).

The structures of the skeletons in the apical region of the coral branch were porous. Complex skeletons form a net-like external surface around the axial corallite, while the calyx can be found in the lumen of the axial corallite (Figure 1A). In stark contrast, the calices, which are the skeletal cups the polyps reside within, were relatively smooth and nearly nonporous (Figure 1B). We also observed that the transformation in the skeleton and lumen at the tip of a newly formed branchlet occurred in three steps (Figure 1C–E), which was evidence of the high activity occurring during coral growth. From step one to step three, the morphological characteristics gradually changed from calice-like to axial corallite-like with an advancement of the growth process. During the transformation from a calice to the axial corallite, the skeletons around the lumen became thicker, and complex gastrovascular canal system was formed in the skeleton (Figure 1E). The calyces, which are the signals in the axial corallite of *A. muricata*, were gradually formed from the calice to step two (Figure 1B–D). Until step two, the structures of both the lumen and axial corallite in the new branch were similar to those of the old branch (Figure 1B,D), and the only difference in the new branch between step two and step three was the length of the lumen (Figure 1D,E).

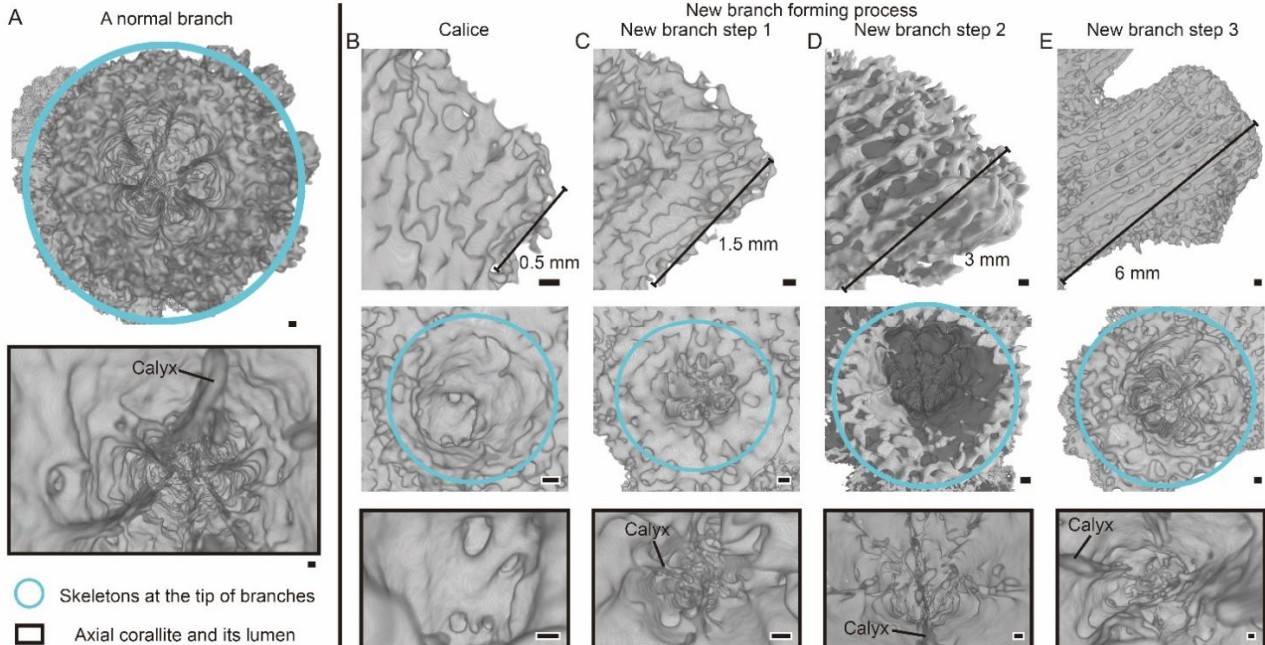

**Figure 1.** Structure of the axial corallite during the processes of new branch formation in *A. muricata*. (**A**) The axial corallite in a healthy and mature coral branch. (**B–E**) A calice transfers into an axial corallite of the new branch. Scale: 0.1 mm.

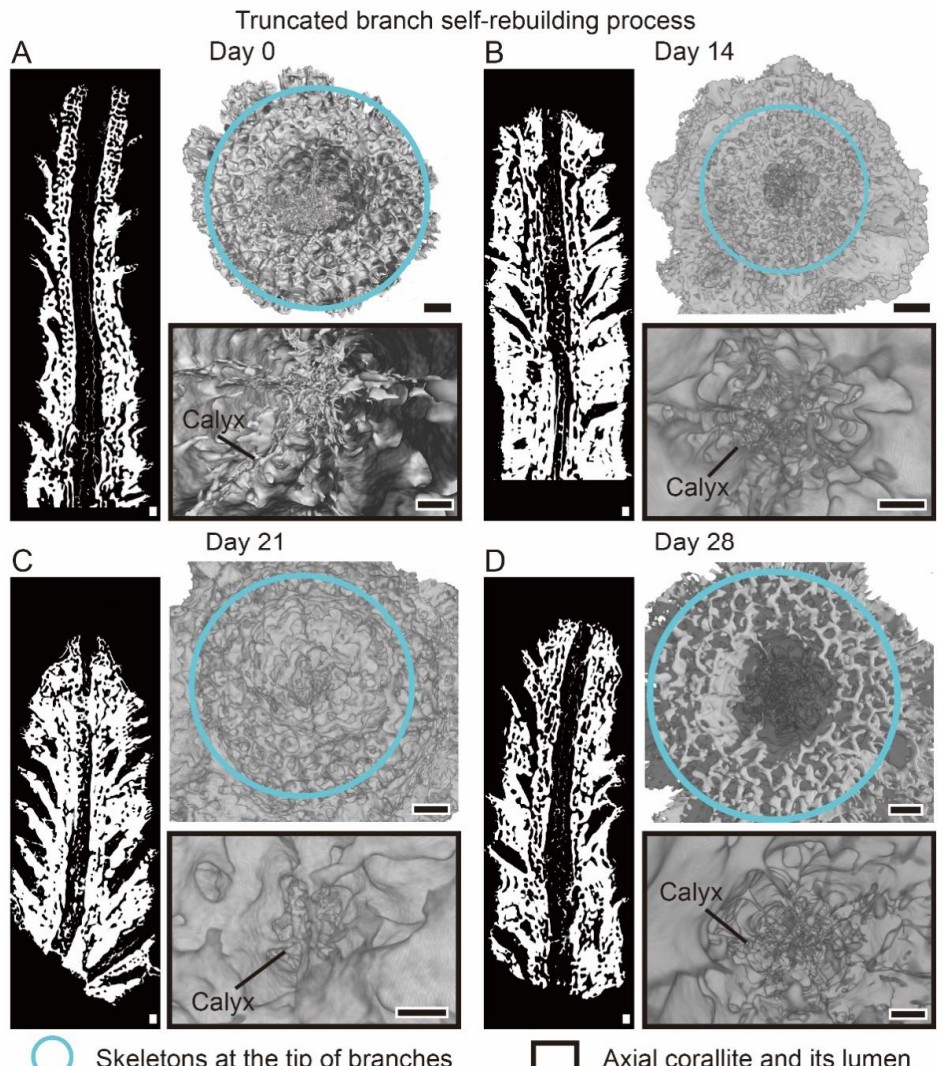

**Figure 2.** Structure of the axial corallite during the process of truncated branch self-rebuilding in *A. muricata*. (**A–D**). The rebuilding process of the truncated branch. Scale: 0.5 mm.

During the tip rebuilding process in truncated branches, a pattern that was not described before appeared in the lumen of the axial corallite (Figure 2A–D). The structure of a truncated branch was similar to a normal branch at day 0, and the only calcareous deposits in the axial corallite were the calyces (Figures 1A and 2A). When the new tip was rebuilt at the truncated area around day 14, irregularly shaped calcareous deposited beneath the tissue onto the axial corallite (Figure 2B). Until day 21, the lumen of the axial corallite in the rebuilt tip was nearly filled with calcareous deposits, and various calcareous deposits even appeared in the lumen below the truncated area (Figure 2C). However, all those sediments disappeared after the rebuilding process at day 28 (Figure 2D), and the shape of the axial corallite returned to the status shown in Figure 1A.

### 3.2. The Changes of Calyx in Axial Corallite Reveal Calcium Transport Patterns

In *A. muricata*, the polyp network is complex because multiple gastrovascular canals are involved. A large number of canals connect all polyps into a holistic network to collaborate in performing biological processes within a single coral colony. To illustrate the patterns in rapid calcium transport, we created 3D reconstructions of the axial corallite and the lumen hidden in coral colonies to obtain information related to calcium carbonate deposition within axial corallite during processes of new branch formation and truncated tip repair (Figures 3–5).

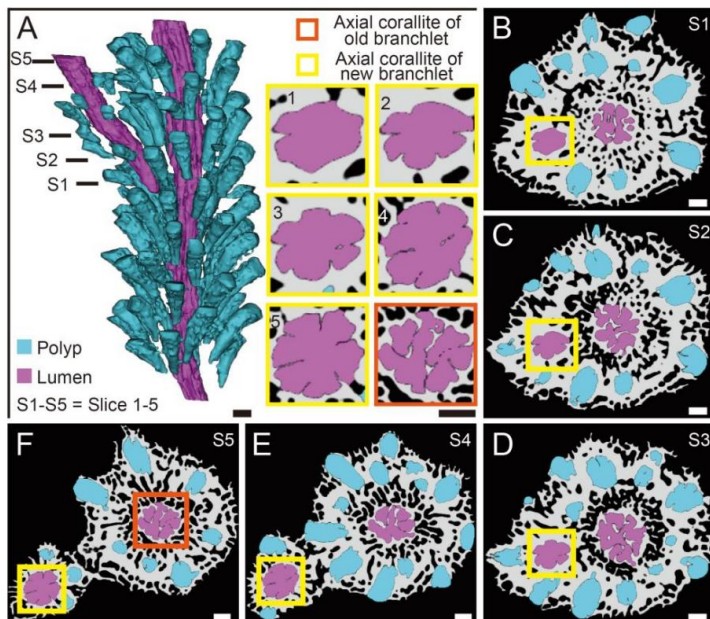

**Figure 3.** Three-dimensional reconstructions visualizing the axial corallite formation during branch-born processes in *A. muricata*. (**A**) An *A. muricata* branch with a newborn branchlet. (**B**–**F**) The S1–S5 cross-sections revealed the calice–axial corallite transformation during the birth of a new branchlet. Scale bars: 1 mm.

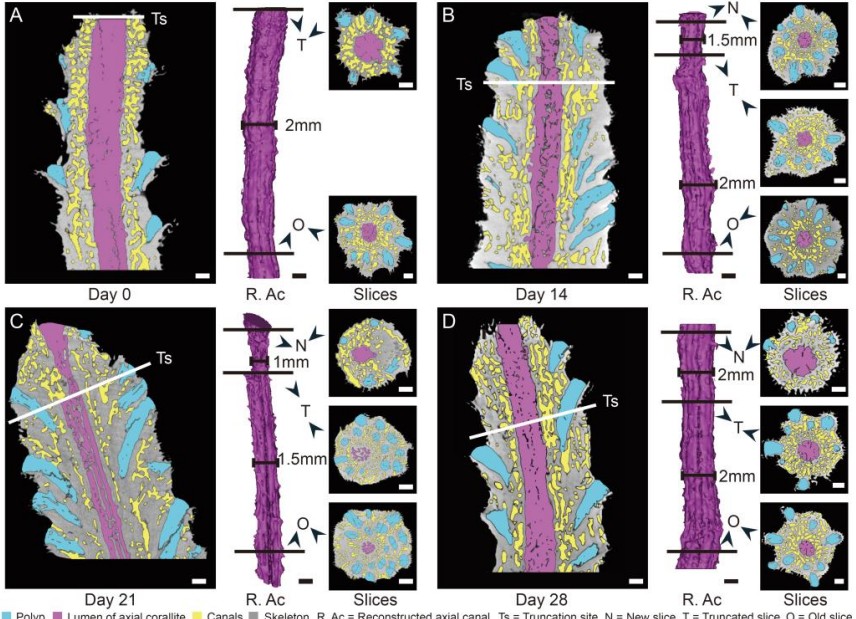

**Figure 4.** Three-dimensional reconstructions revealing calcium transport along the axial canal under the truncated branch rebuilding process in *A. muricata*. (**A**) In 0-day samples, the columnar lumen had a few calcium carbonate deposition scattered inside it, and the cross-sectional diameter of lumen in the axial corallite was about 2.0 mm. (**B**) In 14-day samples, the volume of calcium carbonate deposition within the lumen was increased. The cross-section diameter of the newborn part of the lumen was about 1.5 mm, thinner than the older part. (**C**) In 21-day samples, the calcium carbonate deposition increased and occupied nearly half of the lumen. The cross-section diameter of the lumen was about 1.0 mm in the newborn part, and 1.5 mm in the older part. (**D**) In 28-day samples, there were fewer calcium carbonate depositions left in the axial canal, and the cross-sectional diameter of the lumen was similar to the day 0 group. Scale bars: 1 mm.

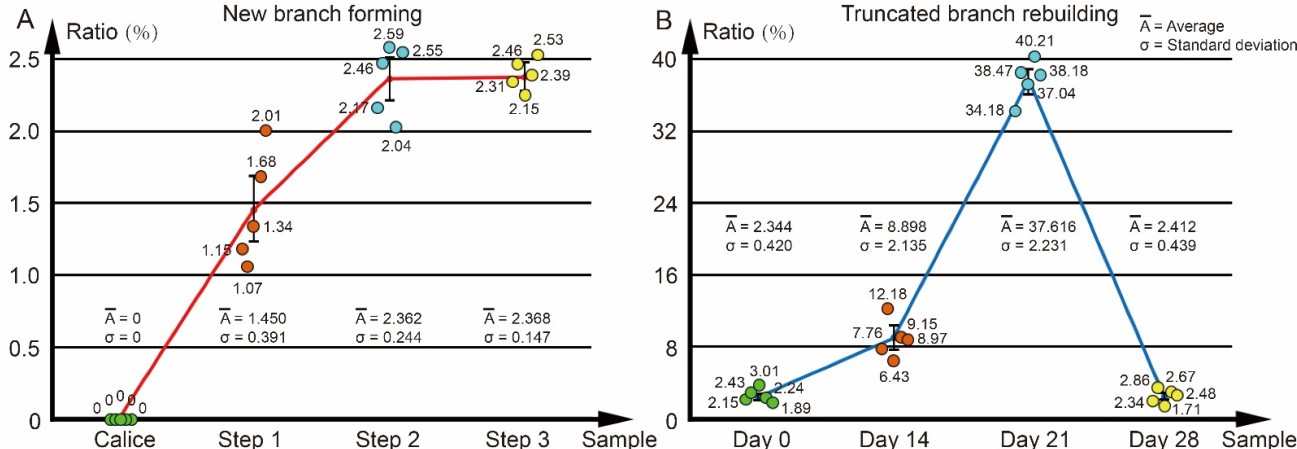

**Figure 5.** Volume ratio of calcium carbonate deposition in the lumen of axial corallite under different growth processes. (**A**) The volume ratio changed during the new branch formation. Green: sample at status of Calice; Red: sample at status of Step 1; Blue: sample at status of Step 2; Yellow: sample at status of Step 3. (**B**) The volume ratio changed during the truncated tip repair. Green: sample at Day 0; Red: sample at Day 14; Blue: sample at Day 21; Yellow: sample at Day 28.

In the *A. muricata* branch, the distance among adjacent calices was similar, and the calices circled the axial corallite along the growth direction. The distances from each calice bottom to the axial corallite were also similar, and complex gastrovascular canals linked the polyps in them (Figure 3). When an *A. muricata* colony branches, the axial canal reveals the branch growth direction, and the new axial corallite appears in the center of the newborn branchlet (Figure 3A). The newly formed axial corallite first appeared at the stage shown in slice one (Figure 3B). At that time, its cross-section was approximately annular, such as that of a common calice. The distance between the bottom of the new axial corallite and that of the old axial corallite was approximately 1.5 mm, and the distance to adjacent calices was approximately 2 to 3 mm (Figure 3B). The shape, location, and distribution of the axial corallite were similar to those of the calices at this stage. In slice 2, the cross-section of the new axial corallite started to present a hexactinal shape (the deposition of calyces started), such as that of an old axial corallite, and two new calices emerged close to this younger one (Figure 3C). At the stage of slice 3, the cross-section of the new axial corallite further approached a hexactinal shape (Figure 3D), and calcium carbonate deposition appeared inside its lumen. Additionally, the skeleton outline of the new branchlet tended to be patterned. In slice 4, new calices appeared between the new and old axial corallite, while the connection between the newborn branchlet and the old branch consisted of only a piece of skeleton (Figure 3E). Up to slice 5, the new axial corallite was surrounded by more calices, and its cross-sectional shape was the same as that of the old one (Figure 3F). The new branchlet separated from the old one, and axial corallite formation was complete. This pseudotime process from slices one to five showed the transformation of a newborn axial corallite from a calice type to a mature state and the birth pattern of a new branchlet. The metamorphosis from calice to corallite indicated why leading polyps were distributed in *A. muricata* branch tips and suggests the budding process in a new branchlet (Figure 6). Meanwhile, the slices of the axial corallite in Figure 3A also revealed the formation of calyces in the axial corallite during the stages from calice to step two in Figure 1.

We also investigated the rebuilding process of a branch and its axial corallite in *A. muricata* through a truncation experiment and a 3D canal reconstruction (Figure 4). The axial corallite in the day 0 group had a smooth surface with the calyces deposited on it, and the cross-sectional diameter of lumen in the axial corallite was approximately 2.0 mm (Figure 4A). Obvious changes appeared in the structure of the axial corallite beginning at day 14, with the appearance of new calcium carbonate deposition on the axial corallite, over the calyces (Figure 4B). The volume of inner skeletons within the lumen increased, and the surface of the axial corallite became rough with more concave structures, indicating that the

axial canal was just squeezed into a tighter space during the linear growth of axial corallites. The cross-sectional diameter of the newborn part of the lumen was approximately 1.5 mm thinner than its previous diameter (Figure 4B). By day 21, the coral branch entered the peak period of the rebuilding process, and its axial corallite structure had changed the most (Figure 4C). The calcium carbonate deposition had been connected into many long column-like structures over the axial corallite, occupying nearly half of the lumen of the axial corallite. The cross-sectional diameter was approximately 1.0 mm, half of the diameter of the day 0 group. Amazingly, the cross-sectional diameter of the previous lumen reduced to 1.5 mm, suggesting that a long-distance calcium transport was also involved in this rebuilding process [8]. In the 28-day samples, only the original calyces were left in the axial corallite, and the cross-sectional diameter of the lumen was similar to that of the day 0 group (Figure 4D). The surface of the axial corallite was smooth again, indicating that the rebuilding process of the branch was almost complete. This rebuilding process, shown in Figure 4, was a normal growth pattern that received much concern. These related 3D reconstructions suggest that the axial canal plays an important role in calcium transport of the truncated branch rebuilding process in *A. muricata*, implying that the temporary storage and remobilization of calcium carbonate presents on the axial corallite during the self-healing process. Meanwhile, this phenomenon indicates that the gastrovascular canal system in one *A. muricata* branch, represented by the axial canal, can connect the polyps in the branch into a network to regulate the rebuilding process of coral growth.

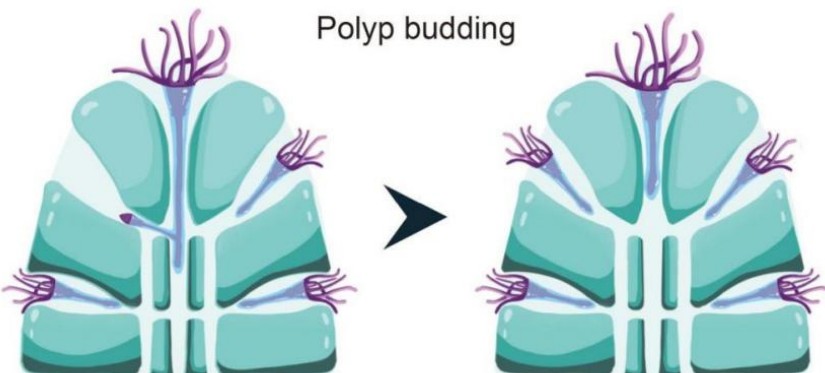

**Figure 6.** Polyp budding schematic according to the canal network reconstructions. The polyp buds after the mineralization of new skeletons, and the newly budded polyp migrates to its new calice through the canals.

To quantify the calcium carbonate deposition in the axial corallite during the two growth processes, we calculated the value of each sample in this study (Figure 5, Method 4.6). No calcium carbonate deposition was found in the calice before its transformation to the axial corallite in the new branch (Figure 5A). During the new branch formation, calcium can be carried along the new axial canal to form its calyx, which is shown as an increase in the volume ratio of calcium carbonate deposition in the lumen of axial corallite (from 0% to 2.362%), between the states of the calice and step two (Figure 5A). The truncated tip repair led to a huge increase (from 2.344% to 37.616%) in the volume ratio of calcium carbonate deposition from day 0 to day 21, and the ratio returned to the initial state after the self-rebuilding (Figure 5B).

## 4. Discussion

### 4.1. The Gastrovascular Canal System Regulates the Budding and Branching Process

In this study, we reconstructed the axial corallite and its lumen through HRCT to investigate the role of the gastrovascular canal system in the budding, branching, calcium transport, and self-healing processes of coral growth (Figures 1–5).

The canal system, which makes up an apparent polyp network, is the basic foundation of coral growth [10,45]. As the skeleton surrounding the largest canal in the network of an *A. muricata* colony, the axial corallite is transformed from specific calices (Figure 1B–E and Figure 3). The chosen calice transforms into the axial corallite of new branch following the regulation of the canal system in the branching process (Figure 3). Meanwhile, the 3D reconstructions revealed that in the newly formed branchlet, new calices distributed near their axial corallite may only obtain polyps from the new axial canal (Figure 3A,E,F), which also suggests the budding patterns in the new branchlet (Figure 6). Although how a coral colony selects a specific calice and induces its transition into a new axial corallite during coral branching is still unclear, the visualization of this process provides a basis for studies in this area.

### 4.2. Calcium Transport along the Axial Canal during Rapid Growth

Calcium can be carried through the gastrovascular canal system toward the growth area of the coral colony [19]. This study revealed that the axial canal plays an important role in calcium transport during two normal growth states: truncated branch self-rebuilding, and new branch formation (Figures 3 and 4). The analysis and speculation regarding calcium transport along the axial canal were mainly based on the visualization and parameter calculations of the calcareous deposits on the wall of the axial corallite, restricting the diameter of the lumen of the axial corallite (Figures 3–5).

In truncated branches, the most active area of calcification was the rebuilding tip at the truncated region and, thus, calcium was delivered to the rebuilding area through the transport system in a colony. A large amount of calcium was transported through the axial canal to the truncated region to rebuild a coral branch, and the increase in calcium content in the axial canal also led to the deposition of calcareous skeletons over calyces, leading to a decrease in the diameter of the lumen of the axial corallite (Figure 4B,C). During the peak period of the branch rebuilding process, in the axial canal, the diameter at the truncated area was reduced by half and the diameter at areas prior to this was reduced by a quarter, while nearly forty percent of the cavity was filled by calcareous deposits (Figures 4C and 5B). After the self-healing of a coral colony, this calcium transportation stops, and the structure of the cavity returns to its initial form (Figure 4D). This self-healing process infers that transport within the gastrovascular system is responsible for supplying the needed materials for deposition to occur, and then dissolution of the calyx that removes calcium carbonate and that restores the original diameter of the axial corallite and axial canal to occur, the removal of calcium carbonate can be used for the linear growth of the skeleton [46]. The data in our work revealed a longer period calcium deposit and release process during the self-healing process, demonstrating that calcium carbonate can be stored temporarily in the calyx and then remobilized as needed for rapid growth.

The formation of a new branchlet during coral branching will also lead to calcium transport in the colony. In this growing process, calcium transport takes place in the axial canal during the deposition of calyces in the axial canal (Figure 1B–D, and Figure 5A). However, this phenomenon did not appear after step 2, which means that with the end of the calyces deposition, the formation of a new branch begins to approach the regular coral growth state, and the calcium transport in the axial canal also tends to stop (Figures 1E and 5A). This indicates that the temporary storage (in the form of calyx) of calcium carbonate in the axial canal may only happen during rapid growth in the coral colony, and the calyx is a form of calcium carbonate stored in the gastrovascular canal system [21]. This truncated branch rebuilding experiment on *A. muricata* also suggests that the polyp network of the gastrovascular canal system makes coral branch growth a kind of integral behavior.

**Author Contributions:** Y.L., C.H., and Z.L. conceived the project. Y.L. wrote the paper and produced the figures. Y.L. reconstructed the images and performed the biological analyses. X.L. collected the coral samples. Y.L., C.H., and Z.L. edited the paper. All authors discussed and commented on the data. All authors have read and agreed to the published version of the manuscript.

**Funding:** This research was funded by the Open Research Fund Program of Guangxi Key Lab. of Mangrove Conservation and Utilization, grant number GKLMC-202002.

**Institutional Review Board Statement:** The study was conducted according to the guidelines of the Declaration of Helsinki, and approved by the Ethics Committee of Institutional Animal Care and Use Committee of NMU (protocol code IACUC-1910003 and date of approval is 10 October 2019).

**Data Availability Statement:** The HRCT data that support the findings of this study are available to share. You may download the HRCT reconstruction data through the following links. *Acropora muricata* colony: doi:10.5061/dryad.wdbrv15nm (https://datadryad.org/stash/share/rWeA0hUxlu__l8sUEdkAIPAtTq3 UGbxsYw095ktjWhs (accessed on 3 March 2021)), *Acropora muricata* branch 1: doi:10.5061/dryad.ghx3ffbnh (https://datadryad.org/stash/share/7YfZPthkA9VP6Djw0OPRyKGhXAP6L-Bb5XYMYVA4ZVQ (accessed on 3 March 2021)), *Acropora muricata* branch 2: doi:10.5061/dryad.p2ngf1vq4 (https://datadryad.org/stash/share/V3iXkB8Fk2M9nvELjGCyOJsRzpNFyBBrKo34oGdwu7M (accessed on 3 March 2021)), *Acropora muricata* branch 3: doi:10.5061/dryad.08kprr524 (https://datadryad.org/stash/share/8SPvv12ifZv_wEf2Uy61PauSkL0oX8tF1iR9lf74DIM (accessed on 3 March 2021)), *Acropora muricata* branch 4: doi:10.5061/dryad.8w9ghx3mb (https://datadryad.org/stash/share/ZTV11TYSlf5 Kgqnfo1UZBdS8KabYJSPrrL7U6jg6Jv0 (accessed on 3 March 2021)), *Acropora muricata* branch 5: doi:10.5061/dryad.pnvx0k6m8 (https://datadryad.org/stash/share/e94f4E80T8ZLfC-YXOACLdMoPMcl4 AfVLwt6xdLvNb8 (accessed on 3 March 2021)).

**Acknowledgments:** We thank the researchers in Guangxi Mangrove Research Center who have assisted us with specimen collection.

**Conflicts of Interest:** The authors declare no conflict of interest.

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
