# Peer review of "Calcium Transport along the Axial Canal in Acropora"

_diversity, doi:10.3390/d13090407_

Round 1

Reviewer 1 Report

The revised manuscript is much improved as the authors have addressed several of my initial key concerns, which included:

  1. Lack of application of knowledge
  2. Lack of background literature
  3. Lack of sufficient methodological detail

The grammar is also improved, BUT there are still several grammatical errors thorough out the document. As a reviewer, it is not my remit to edit for grammatical errors, but to verify the science and application.  Therefore, I strongly recommend the authors to get someone with a strong English background to edit and review the manuscript carefully before publication.

Author Response

Point 1: The grammar is also improved, BUT there are still several grammatical errors thorough out the document. As a reviewer, it is not my remit to edit for grammatical errors, but to verify the science and application. Therefore, I strongly recommend the authors to get someone with a strong English background to edit and review the manuscript carefully before publication.

[Response 1] Thanks for your valuable comments, and we sought help from the English editing service of MDPI, they helped us to correct our grammatical errors. We also got help from a biologist to correct the terminology mistakes in this manuscript. According to their advice, we did following changes:

(Lines 10-11) We changed “In Acropora, the complex canals in a coral colony connect all polyps into a holistic network to collaborate in performing biological processes.” to “In Acropora, the complex canals in a coral colony connect all polyps to a holistic network, enabling them to collaborate in performing biological processes.”

(Lines 114-115) We changed “We also weekly added minerals to the tank” to “We also added minerals to the tank weekly”.

We changed “iconic hexactin skeletons” to “calyx” or “calyces” (according to singular and plural).

We used “gastrovascular canal system” to refer the complex calcium transport system (according to references 21-23 and 42).

We used “temporarily storage”, “remobilization” to describe the calcium transport patterns shown in this work (according to references 21 and 54).

Reviewer 2 Report

The results of this study are very interesting and do shed some light on control of calcium carbonate deposition and growth of the axial corallite in Acropora. These results should be published, as they demonstrate that calcium carbonate can be stored temporarily and then remobilized as need for rapid growth. However, these results are presented as if there is no prior information about growth in Acropora. In fact, there have been a number of studies, which have determined spatial and temporal patterns of calcium carbonate accretion in Acropora.

 The authors seem to be confusing the gastrovascular canal system (lined by endoderm (gastroderm) and underlain by calicoblastic ectoderm) with the skeleton, which is deposited beneath the tissue. The gastrovascular canal system consists of the axial canal lying within the axial corallite, radial canals in the lateral corallites, and a network of smaller diameter canals connecting these together. The movement of fluid within the gastrovascular system can bring materials to different parts of the colony as required (e.g., Pearse VB, Muscatine L (1971) Role of symbiotic algae (zooxanthellae) in coral calcification. Biol Bull 141:350–363)

The present study employs a method that only presents what has occurred within the skeleton of the axial corallite. It examines regrowth of the axial corallite when the branch tip is broken, and also formation of a new axial skeleton as a new branch forms. It probably correctly infers that transport within the gastrovascular system is responsible for supplying the needed materials for deposition to occur, and then dissolution of the calyx that removes calcium carbonate and that restores the original diameter of the axial corallite and axial canal to occur.

There are two papers that describe temporal and spatial patterns of calcium carbonate accretion in Acropora cervicornis (a staghorn coral, similar to A. muricata, but from the Caribbean):

Gladfelter EH (1982) Skeletal development in Acropora cervicornis I. Patterns of calcium carbonate accretion in the axial corallite. Coral Reefs 1: 45-51

Gladfelter EH (1983) Skeletal development in Acropora cervicornis II. Diel patterns of calcium carbonate accretion. Coral Reefs 2: 91-100

These articles discuss diel cycle as well as decadal cycles of carbonate deposition in the axial corallite.

The authors should discuss what is known about temporal cycles in calcium carbonate deposition. They describe a pattern of deposition and dissolution that occurs over approximately a month period. However, the methods were unclear about when during the day the sampling occurred (the same time each day?). The methods also omitted information about the light regime (light levels, hours of dark/light) during the experimental period. 

The first obvious pattern of calcium carbonate deposition is the diel cycle. There have been multiple studies about this, but perhaps the most pointed description of calcification in staghorn Acropora is:

“It seems possible that the symbiotic association permits rapid growth because the coral can invest in flimsy scaffolding at night with the certainty that bricks and mortar will be available in the morning” (Barnes DJ, Crossland CJ (1980) Diurnal and seasonal variations in the growth of staghorn coral measured by time-lapse photography. Limnol Oceanogr 25: 1113-1117)

Decadal patterns of deposition in the axial corallite in Acropora were demonstrated (see Figs. 5 and 6) in the article by Gladfelter (1982) referenced above. The gastrovascular canals ramify throughout the branch and over the decades the axial corallite fills in. It may be that this “stored” calcium carbonate can be remobilized and then redeposited in growth axial corallites as describe in the current article.

The authors should consider placing there results in a different context. There is no rationale for how these results are related to coral restoration work. If that is the framework that the authors chose to place these results, they need to make a more clear connection. However, these are not really “extreme growth events” as the authors have described them, but rather a normal pattern that has not been describe before. It does illustrate that the calcium carbonate within the skeleton can be remobilized as needed and that is a significant result.

The term “iconic hexactin skeleton” is unfamiliar and unclear to me.

Author Response

1) Background:

Point 1: The results of this study are very interesting and do shed some light on control of calcium carbonate deposition and growth of the axial corallite in Acropora. These results should be published, as they demonstrate that calcium carbonate can be stored temporarily and then remobilized as need for rapid growth. However, these results are presented as if there is no prior information about growth in Acropora. In fact, there have been a number of studies, which have determined spatial and temporal patterns of calcium carbonate accretion in Acropora.

The present study employs a method that only presents what has occurred within the skeleton of the axial corallite. It examines regrowth of the axial corallite when the branch tip is broken, and also formation of a new axial skeleton as a new branch forms. It probably correctly infers that transport within the gastrovascular system is responsible for supplying the needed materials for deposition to occur, and then dissolution of the calyx that removes calcium carbonate and that restores the original diameter of the axial corallite and axial canal to occur.

There are two papers that describe temporal and spatial patterns of calcium carbonate accretion in Acropora cervicornis (a staghorn coral, similar to A. muricata, but from the Caribbean):

Gladfelter EH (1982) Skeletal development in Acropora cervicornis I. Patterns of calcium carbonate accretion in the axial corallite. Coral Reefs 1: 45-51

Gladfelter EH (1983) Skeletal development in Acropora cervicornis II. Diel patterns of calcium carbonate accretion. Coral Reefs 2: 91-100

These articles discuss diel cycle as well as decadal cycles of carbonate deposition in the axial corallite.

[Response 1] Thanks for your valuable comments, we have read the references you listed and rewrote the background in the abstract and introductions:

(Lines 13-16) We changed “However, few studies have considered the regulation of calcium transport in Acropora.” to “Former researches determined spatial and temporal patterns of calcium carbonate accretion in Acropora through scanning electron microscopy, however, the patterns of calcium transport in the axial canal remain obscure.”

(Lines 45-53) We changed “the axial canal is unique, being the largest canal along the branch center in an Acropora colony, and its extension reveals the growth directions of the coral branch (8,9). However, the role of the axial canal in an Acropora colony and details of its structural transformation during coral growth remain obscure” to “the gastrovascular canal system consists of the axial canal lying within the axial corallite, radial canals in the lateral corallites, and a network of smaller diameter canals connecting these together (21). The movement of fluid within the gastrovascular system can bring materials to different parts of the colony as required (22). Former researches described temporal and spatial patterns of calcium carbonate accretion in Acropora cervicornis, and discussed diel cycle as well as decadal cycles of carbonate deposition in the axial corallite (21,23). However, the patterns of calcium transport in the axial canal and details of its structural transformation during coral growth remain obscure”.

(Lines 57-58) We added “Errors may occur in the measurements, because of the requirement to remove tissue from the skeletons (30).”

(Line 62) We changed “how the axial canal participates in coral growth and calcium transport (27-29)” to “the patterns of calcium transport in the axial canal (31-33)”.

(Lines 414-418, 431-432) We added references 21-23, 30.

2) Methods:

Point 2: The authors should discuss what is known about temporal cycles in calcium carbonate deposition. They describe a pattern of deposition and dissolution that occurs over approximately a month period. However, the methods were unclear about when during the day the sampling occurred (the same time each day?). The methods also omitted information about the light regime (light levels, hours of dark/light) during the experimental period.

[Response 2] We added these information to our methods:

(Lines 147-150) “The sampling occurred the same time each day, at 12 o’clock noon. The illuminance in the tank was between 600-800 PAR(µmol/m²·s), and the time of irradiation was between 8 AM to 8 PM.”

3) Research design and the discussions:

Point 3: The first obvious pattern of calcium carbonate deposition is the diel cycle. There have been multiple studies about this, but perhaps the most pointed description of calcification in staghorn Acropora is:

“It seems possible that the symbiotic association permits rapid growth because the coral can invest in flimsy scaffolding at night with the certainty that bricks and mortar will be available in the morning” (Barnes DJ, Crossland CJ (1980) Diurnal and seasonal variations in the growth of staghorn coral measured by time-lapse photography. Limnol Oceanogr 25: 1113-1117)

Decadal patterns of deposition in the axial corallite in Acropora were demonstrated (see Figs. 5 and 6) in the article by Gladfelter (1982) referenced above. The gastrovascular canals ramify throughout the branch and over the decades the axial corallite fills in. It may be that this “stored” calcium carbonate can be remobilized and then redeposited in growth axial corallites as describe in the current article.

The authors should consider placing there results in a different context. There is no rationale for how these results are related to coral restoration work. If that is the framework that the authors chose to place these results, they need to make a more clear connection.

[Response 3] We read the references you listed and replaced our results to a different context according to your suggestions. Instead of the coral reef restoration projects, we focused on the studies about calcium transport patterns in the gastrovascular system of Acropora colonies:

(Lines 23-26) We changed “This work indicates that the axial canal can transport calcium to form hexactin skeletons in a new branch and rebuild the tip of a truncated branch. The calcium transport along canal network regulates various growth processes, including budding, branching, skeleton forming, and self-rebuilding of an Acropora colony. Understanding the changes in canal function under normal and extreme conditions will provide theoretical guidance for restoration and protection of coral reefs.” to “This work demonstrates that calcium carbonate can be stored temporarily and then remobilized as need for rapid growth. The results of this study shed light on control of calcium carbonate deposition and growth of the axial corallite in Acropora.”

(Lines 58-91) We changed “This study expands our understanding of calcium transport patterns in Acropora, which is conducive to the breeding and protection of reef-building corals (38-41). In addition, studying the internal structure and function of coral canals provides theoretical guidance for the current research into constructing artificial reefs (42,43) and reconstructing coral reefs through 3D printing (44-47), laying a scientific foundation for the establishment of an improved coral reef restoration system (41, 48-50).” to “This study expands our understanding of calcium transport patterns in Acropora, which sheds light on control of calcium carbonate deposition and growth of the axial corallite in Acropora (21-23, 42). The data enable us to inference the biochemical mechanisms of calcium movement in corals (43). In addition, calcification processes are also important geochemically (44,45), and our work provides theoretical guidance for the current research into the active calcium transport (46-48), laying a scientific foundation for the establishment of an improved coral calcification model (49-51).”

(Lines 275-276) We changed “calcium transport” to “the transportation, temporary storage, and remobilization of calcium carbonate”

(Lines 334-343) We added “This self-healing process infers that transport within the gastrovascular system is responsible for supplying the needed materials for deposition to occur, and then dissolution of the calyx that removes calcium carbonate and that restores the original diameter of the axial corallite and axial canal to occur. In former studies, Barnes, D. J. and Crossland, C. J. found that the symbiotic association of zooxanthellae and coral permits rapid growth because the coral can invest in flimsy scaffolding at night with certainty that the bricks and mortar will be available in the morning (54). The data in our work reveals a longer-period calcium deposit and release process during the self-healing process, demonstrating that calcium carbonate can be stored temporarily in calyx and then remobilized as need for rapid growth.” after “After the self-healing ... its initial form (Fig. 4D).”

(Lines 350-353) We changed “This indicates that the calcium transportation through the axial canal may only happen during the self-healing and branching processes in the coral colony, suggesting further study is required.” to “This indicates that the temporary storage (in the form of calyx) of calcium carbonate in the axial canal may only happen during rapid growth in the coral colony, and the calyx is a form of calcium carbonate stored in the gastrovascular canal system (21).” 

(Lines 455-470, 475-476) We added references 42-51, 54.

4) Terminology:

Point 4: The authors seem to be confusing the gastrovascular canal system (lined by endoderm (gastroderm) and underlain by calicoblastic ectoderm) with the skeleton, which is deposited beneath the tissue. The gastrovascular canal system consists of the axial canal lying within the axial corallite, radial canals in the lateral corallites, and a network of smaller diameter canals connecting these together. The movement of fluid within the gastrovascular system can bring materials to different parts of the colony as required (e.g., Pearse VB, Muscatine L (1971) Role of symbiotic algae (zooxanthellae) in coral calcification. Biol Bull 141:350–363)

The term “iconic hexactin skeleton” is unfamiliar and unclear to me.

[Response 4] We corrected the terminology mistakes according your suggestions and the references you listed for us:

We changed “iconic hexactin skeletons” to “calyx” or “calyces”, and marked the position of calyx in Figure 1 and 2 (Lines 167 and 197).

We used “gastrovascular canal system” to refer the complex calcium transport system.

We used “temporarily storage”, “remobilization” to describe the calcium transport patterns shown in this work.

Round 2

Reviewer 2 Report

This is much improved, however, I still get the sense that the authors are confusing the axial (and other) canals  (which are living tissue) with the corallites (and other skeletal elements) that the canals lie within The interconnected tissue is topologically outside the skeleton that it deposits (and, as this study demonstrated, also resorbs). This article still needs another revision, but the findings are very significant and should be published.

When there is linear growth of the axial corallite, either due to breakage of the tip, or in the development of a new branch (thus a new axial corallite), calcium is apparently transported within the axial canal, but then deposited beneath the tissue onto the skeleton that is the axial corallite, and then resorbed to contribute to the linear extension of the corallite.

13-15

I’d leave this sentence out. The present study has shown yet another pattern of calcium carbonate deposition…presumably all the patterns shown have been dependent to some degree on transport of calcium within the axial canal.

16 This study demonstrated a change in the corallite NOT the axial canal (for all we know the axial canal was just squeezed into a tighter space during the linear growth of axial corallites, and then resumed its “normal” configuration).

17 What was observed is patterns calcium carbonate deposition within  axial corallite during processes of new branch formation and truncated tip repair.

What I think is very exciting about this study is the potential for a much more detailed study of the calicoblastic tissue during the buildup of calcium carbonate and conversely during the dissolution of the axial corallite in the 28 day period. This is ripe for a TEM study of the tissue, and could contribute greatly to an understanding of the process of calcification.

34 again there is confusion between the tissue and the skeleton. The canals are within the tissue (ultimately connected to the outside via the mouths of the polyps). The tissue gastrovascular canals lie within lumen of the skeleton. The calices are the skeletal cups the polyps reside within.

“The resulting skeleton is highly porous with all surfaces covered by the continuous calicoblastic epithelium. This cell layer is separated by thin mesoglea from the flagellated gastrodermis which lines the highly ramified coelenteron.” Gladfelter 1982

60 the patterns of calcium transport in the axial canal [evidenced in changes in the diameter of the axial corallite]

78 and 79 and 80 the axial corallite NOT the axial canal

84 I am not clear how this “enables us to inference biochemical mechanisms of calcium movement in corals…Referenced article #49, first author is Goreau

149 and 152 and 155, 156, 168 axial corallite not axial canal

168 the skeletal protrusions are part of the axial corallite…the inner wall

A distinction needs to be clear between what is happening in the skeleton (the axial corallite and other corallites) and the axial canals which are tissue (and was not investigated in this study)…168-298

312 calcareous deposites on the wall of the axial corallite, restricting the diameter of the lumen of the corallite

307 and 316…what is meant by hydroplasm? Complete cells can be transported in the gastrovascular canals of hydroids…

312 should read in the lumen of of the axial corallite

321 axial canal cavity = lumen of the axial corallite

329 add that the removal of calcium carbonate can be used for linear growth of the skeleton

335 Nobody has looked at the lumen of the axial canal on a diel basis, maybe stored material in the day is used to fuel the linear growth at night

Author Response

Response to Reviewer 2 Comments

1) Point 1: This is much improved, however, I still get the sense that the authors are confusing the axial (and other) canals (which are living tissue) with the corallites (and other skeletal elements) that the canals lie within The interconnected tissue is topologically outside the skeleton that it deposits (and, as this study demonstrated, also resorbs).

[Response 1] Thanks for your valuable comments, we have corrected these confusions according to your explanation. And the details are shown in the following point to point revisions.

2) Point 2: Id leave this sentence out. The present study has shown yet another pattern of calcium carbonate deposition...presumably all the patterns shown have been dependent to some degree on transport of calcium within the axial canal.

[Response 2] (Line 13) We were agree with the editor and deleted the sentence “Former researches determined spatial and temporal patterns of calcium carbonate accretion in Acropora through scanning electron microscopy, however, the patterns of calcium transport in the axial canal remain obscure”. 

3) Point 3: This study demonstrated a change in the corallite NOT the axial canal (for all we know the axial canal was just squeezed into a tighter space during the linear growth of axial corallites, and then resumed its “normal” configuration).

[Response 3] (Line 14) We changed “of the axial canal in” to “in the corallite of”.

4) Point 4: What was observed is patterns calcium carbonate deposition within  axial corallite during processes of new branch formation and truncated tip repair.

[Response 4] (Lines 15-16) We changed “the axial canal during the processes of new branch formation and truncated branch rebuilding” to “calcium carbonate deposition within axial corallite during processes of new branch formation and truncated tip repair”.

5) Point 5: again there is confusion between the tissue and the skeleton. The canals are within the tissue (ultimately connected to the outside via the mouths of the polyps). The tissue gastrovascular canals lie within lumen of the skeleton. The calices are the skeletal cups the polyps reside within.

“The resulting skeleton is highly porous with all surfaces covered by the continuous calicoblastic epithelium. This cell layer is separated by thin mesoglea from the flagellated gastrodermis which lines the highly ramified coelenteron.” Gladfelter 1982

[Response 5] (Lines 33-35) We changed “Various types of canals, including calices, axial canals, and other internal canals, support the canal network in an Acropora colony and, thus, the physiological processes of coral growth” to “The tissue gastrovascular canals lie within lumen of the skeleton in an Acropora colony, and materials can be transported within the canals”.

6) Point 6: the patterns of calcium transport in the axial canal [evidenced in changes in the diameter of the axial corallite]

[Response 6] (Lines 58-59) We changed “evidence about the patterns of calcium transport in the axial canal” to “evidence in changes in the diameter of the axial corallite”.

7) Point 7: 78 and 79 and 80 the axial corallite NOT the axial canal

[Response 7] (Lines 77-79) We changed the three “axial canal” to “axial corallite”.

8) Point 8: I am not clear how this “enables us to inference biochemical mechanisms of calcium movement in corals…Referenced article #49, first author is Goreau

What I think is very exciting about this study is the potential for a much more detailed study of the calicoblastic tissue during the buildup of calcium carbonate and conversely during the dissolution of the axial corallite in the 28 day period. This is ripe for a TEM study of the tissue, and could contribute greatly to an understanding of the process of calcification.

[Response 8] (Lines 84-88) We changed “The data enable us to inference the biochemical mechanisms of calcium movement in corals (43). In addition, calcification processes are also important geochemically (44,45), and our work provides theoretical guidance for the current research into the active calcium transport (46-48), laying a scientific foundation for the establishment of an improved coral calcification model (49-51)” to “The data enable a much more detailed study of the calicoblastic tissue during the buildup of calcium carbonate and conversely during the dissolution of the axial corallite in the 28 day period. This is ripe for a transmission electron microscope (TEM) study of the tissue, and could contribute greatly to an understanding of the process of calcification (43)”.

(Line 451) We also deleted references 44-51 and corrected the errors in the reference 43.

9) Point 9: 149 and 152 and 155, 156, 168 axial corallite not axial canal

[Response 9] (Lines 159-169) We changed these “axial canal” to “axial corallite”.

10) Point 10: the skeletal protrusions are part of the axial corallite…the inner wall

[Response 10] (Lines 169-170) We changed “while calyx (skeletal protrusions) can be found in the cavity of the axial canal” to “while calyx can be found in the lumen of the axial corallite”.

11) Point 11: A distinction needs to be clear between what is happening in the skeleton (the axial corallite and other corallites) and the axial canals which are tissue (and was not investigated in this study)

axial canal cavity = lumen of the axial corallite

[Response 11] (Lines 170-308, 324) We corrected the misapplication of the phrases like “axial canals”, “axial canal cavity” and a few other description mistakes in the results and discussions, and changed them into the correct use of “axial corallite”, “lumen of the axial corallite”, et al.

We also corrected the captions of Figure 1-4, and the figure legends of Figure 1-6.

12) Point 12: calcareous deposites on the wall of the axial corallite, restricting the diameter of the lumen of the corallite

should read in the lumen of the axial corallite

[Response 12] (Lines 317-318) We changed “in the canal cavity” to “on the wall of the axial corallite, restricting the diameter of the lumen of the axial corallite”.

13) Point 13: what is meant by hydroplasm? Complete cells can be transported in the gastrovascular canals of hydroids...

[Response 13] (Lines 313, 321) We misunderstood the descriptions in reference 20, and found that these two sentences “According to previous studies of the transport system in Acropora branches, the axial canal is not involved in the transport of hydroplasm (20). However”, “Although the axial canal does not play a major role in the transport of hydroplasm in Acropora, growing process, like truncated branch rebuilding, may change the role of the axial canal in the transport system of A. muricata (Fig. 2, Fig. 4)” should be deleted.

14) Point 14: add that the removal of calcium carbonate can be used for linear growth of the skeleton

[Response 14] (Lines 333-334) We added “the removal of calcium carbonate can be used for linear growth of the skeleton”.

15) Point 14: Nobody has looked at the lumen of the axial canal on a diel basis, maybe stored material in the day is used to fuel the linear growth at night

[Response 15] (Line 334) We deleted “In former studies, Barnes, D. J. and Crossland, C. J. found that the symbiotic association of zooxanthellae and coral permits rapid growth because the coral can invest in flimsy scaffolding at night with certainty that the bricks and mortar will be available in the morning”.

This manuscript is a resubmission of an earlier submission. The following is a list of the peer review reports and author responses from that submission.

Round 1

Reviewer 1 Report

The manuscript entitled ‘Axial canal regulates the processes of coral branching and calcareous transportation in Acropora’ provides a detailed assessment on how the axial canal may be involved in important processes such as growth and rebuilding of branches.  I do, however, have some concerns with the paper, which include: 1) the lack of a detailed review of the literature (only 10 references cited) and very limited placement of the value of this research in a broader context i.e. how is this knowledge applied?, 2) the fact that this study is very similar to that of a previous study by Yixin Li on Pocillopora sp.  Do the authors intend to produce a suite of papers using the same technique on a range of coral species with limited regard to the usefulness of the new knowledge of coral building processes?, and 3) the number of grammatical errors throughout the document.  I have picked up on several, but really it needs a thorough check to ensure the English is of a higher standard.  Given the lack of ‘bigger picture context’ provided for the paper, the lack of novelty of the technique used (as already published) and the grammatical issues, I recommend the paper not be published.  What would be really interesting, and far more useful from an applied angle, is to collect coral samples from different reef environments (e.g. high/low light/nutrients/wave energy) and access how these axial regulation processes are influenced by environmental conditions e.g. under what conditions do these processes act more or less efficiently. This knowledge could be very useful for e.g. coral reef restoration.

More detailed comments below:

Line 10; remove comma after that

Line 10: change ‘and distributes’ to positioned

Line 15-16: speculative so language should be toned down

Line 17: skeleton isn’t transported!  Change skeleton to something else e.g. material

Line 18: change to ‘represented as a change’, and remove from ‘and calcareous…….’

Line 19-20: whats the value of this research? Bigger picture??

Line 25: change to ‘Coral reefs are highly diverse ecosystems….’

Line 31: change processes in coral growth to processes of coral growth

Line 32: change to ‘the axial canal is unique being the largest canal…’

Line 33: change to ‘of the coral branch’

Line 35: change to ‘canal does not play..’

Line 39: change to ‘of coral colonies’

Line 40: change to ‘how the axial canal’

Line 46: incorrect – by studying how a coral colony grows does NOT give information on how a reef grows!  Very different!!

Line 47: change to ‘the regulatory processes of the axial canal’

Line 58 -59 : should be part of the figure caption not the main text

Line 106 -108: this all seems very speculative and should be written as such

Line 137 to 141: very difficult to read.  Needs to be broken down into smaller sentences and re-written

Figure 4b caption should read as 14 not 21 days.

Line 172: change to ‘deliver to the rebuilding’

Line 176: change to ‘rebuild the coral branch’

Line 178: change to ‘represented as a’

Line 179: change to ‘of the coral colony’

Line 180: change to ‘the deposited calcareous skeleton reduces leading to a decrease’

Line 181: change to ‘canal. At this stage, the structure of the cavity’

Line 185: change to ‘happen during the self-healing process in the coral colony suggesting further study is required.’

Line 186; change to ‘also suggests’

Ethics: surely you need to include an ethics approval number for verification

Line 195: last time I checked the south china sea was a pretty large place!  Can you be a lot more specific as to the location of sample collection, and include details of the type of reef, depth of site collection, environmental conditions at site – all key info needed.

Line 198 -199: how big was the colony and branches used?  Did the branches come from different colonies?  Where they genetically different? Is this important?  Lacking important details here

Line 201 to 208:  tense switch from past to present.  Choose one a stick to it.

Line 219 -237: this is tedious to read as written, yet important information to include.  This would be much better of in a table for easier access of information, with a brief accompanying summary text.

Line 245: change to ‘We selected four groups of A.muricata branches of similar size (X cm) and shape (XXX)’ and fill in the XXXX blanks

Line 246: change to ‘same environments. The truncated samples were assessed using HRCT at day 0…’